# META-REINFORCEMENT LEARNING ROBUST TO DISTRIBUTIONAL SHIFT VIA MODEL IDENTIFICATION AND EXPERIENCE RELABELING

## ABSTRACT

Reinforcement learning algorithms can acquire policies for complex tasks autonomously. However, the number of samples required to learn a diverse set of skills can be prohibitively large. While meta-reinforcement learning methods have enabled agents to leverage prior experience to adapt quickly to new tasks, their performance depends crucially on how close the new task is to the previously experienced tasks. Current approaches are either not able to extrapolate well, or can do so at the expense of requiring extremely large amounts of data for on-policy meta-training. In this work, we present model identification and experience relabeling (MIER), a meta-reinforcement learning algorithm that is both efficient and extrapolates well when faced with out-of-distribution tasks at test time. Our method is based on a simple insight: we recognize that dynamics models can be adapted efficiently and consistently with off-policy data, more easily than policies and value functions. These dynamics models can then be used to continue training policies and value functions for out-of-distribution tasks without using meta-reinforcement learning at all, by generating synthetic experience for the new task.

## 1 INTRODUCTION

Recent advances in reinforcement learning (RL) have enabled agents to autonomously acquire policies for complex tasks, particularly when combined with high-capacity representations such as neural networks (Lillicrap et al., 2015; Schulman et al., 2015; Mnih et al., 2015; Levine et al., 2016). However, the number of samples required to learn these tasks is often very large. Meta-reinforcement learning (meta-RL) algorithms can alleviate this problem by leveraging experience from previously seen related tasks (Duan et al., 2016; Wang et al., 2016; Finn et al., 2017a), but the performance of these methods on new tasks depends crucially on how close these tasks are to the meta-training task distribution. Meta-trained agents can adapt quickly to tasks that are similar to those seen during meta-training, but lose much of their benefit when adapting to tasks that are too far away from the meta-training set. This places a significant burden on the user to carefully construct meta-training task distributions that sufficiently cover the kinds of tasks that may be encountered at test time.

Many meta-RL methods either utilize a variant of model-agnostic meta-learning (MAML) and adapt to new tasks with gradient descent (Finn et al., 2017a; Rothfuss et al., 2018; Nagabandi et al., 2018), or use an encoder-based formulation that adapt by encoding experience with recurrent models (Duan et al., 2016; Wang et al., 2016), attention mechanisms (Mishra et al., 2017) or variational inference (Rakelly et al., 2019). The encoder-based methods struggle when adapting to out-of-distribution tasks, because the adaptation procedure is entirely learned and carries no guarantees for out-of-distribution inputs (as with any learned model). Methods that utilize gradient-based adaptation have the potential of handling out-of-distribution tasks more effectively, since gradient descent corresponds to a well-defined and consistent learning process that has a guarantee of improvement regardless of the task (Finn & Levine, 2018). However, in the RL setting, these methods (Finn et al., 2017a; Rothfuss et al., 2018) utilize on-policy policy gradient methods for meta-training, which require a very large number meta-training samples (Rakelly et al., 2019).

In this paper, we aim to develop a meta-RL algorithm that can both adapt effectively to out-of-distribution tasks and be meta-trained efficiently via off-policy value-based algorithms. One approach

Figure 1: Overview of our approach. The model context variable ($\phi$) is adapted using gradient descent, and the adapted context variable ($\phi_\mathcal{T}$) is fed to the policy alongside state so the policy can be trained with standard RL (Model Identification). The adapted model is used to relabel the data from other tasks by predicting next state and reward, generating synthetic experience to continue improving the policy (Experience Relabeling).

might be to directly develop a value-based off-policy meta-RL method that uses gradient-based meta-learning. However, this is very difficult, since the fixed point iteration used in value-based RL algorithms does not correspond to gradient descent, and to our knowledge no prior method has successfully adapted MAML to the off-policy value-based setting. We further discuss this difficulty in Appendix A. Instead, we propose to leverage a simple insight: *dynamics and reward models* can be adapted consistently, using gradient based update rules with off-policy data, even if policies and value functions cannot. These models can then be used to train policies for out-of-distribution tasks without using meta-RL at all, by generating synthetic experience for the new tasks.

Based on this observation, we propose **model identification and experience relabeling (MIER)**, a meta-RL algorithm that makes use of two independent novel concepts: **model identification** and **experience relabeling**. Model identification refers to the process of identifying a particular task from a distribution of tasks, which requires determining its transition dynamics and reward function. We use a gradient-based *supervised* meta-learning method to learn a dynamics and reward model and a (latent) *model context variable* such that the model quickly adapts to new tasks after a few steps of gradient descent on the context variable. The context variable must contain sufficient information about the task to accurately predict dynamics and rewards. The policy can then be conditioned on this context (Schaul et al., 2015; Kaelbling, 1993) and therefore does *not* need to be meta-trained or adapted. Hence it can be learned with any standard RL algorithm, avoiding the complexity of meta-reinforcement learning. We illustrate the model identification process Figure 1 (left).

When adapting to out-of-distribution tasks at meta-test time, the adapted context variable may itself be out of distribution, and the context-conditioned policy might perform poorly. However, since MIER adapts the model with gradient descent, we can continue to improve the model using more gradient steps. To continue improving the policy, we leverage all data collected from other tasks during meta-training, by using the learned model to *relabel* the next state and reward on every previously seen transition, obtaining synthetic data to continue training the policy. We call this process, shown in the right part of Figure 1, **experience relabeling**. This enables MIER to adapt to tasks outside of the meta-training distribution, outperforming prior meta-reinforcement learning methods in this setting.

## 2 PRELIMINARIES

Formally, the reinforcement learning problem is defined by a Markov decision process (MDP). We adopt the standard definition of an MDP, $\mathcal{T} = (\mathcal{S}, \mathcal{A}, p, \mu_0, r, \gamma)$, where $\mathcal{S}$ is the state space, $\mathcal{A}$ is the action space, $p(\mathbf{s}'|\mathbf{s}, \mathbf{a})$ is the unknown transition probability of reaching state $\mathbf{s}'$ at the next time step when an agent takes action $\mathbf{a}$ at state $\mathbf{s}$, $\mu_0(\mathbf{s})$ is the initial state distribution, $r(\mathbf{s}, \mathbf{a})$ is the reward function, and $\gamma \in (0, 1)$ is the discount factor. An agent acts according to some policy $\pi(\mathbf{a}|\mathbf{s})$ and the learning objective is to maximize the expected return, $\mathbb{E}_{\mathbf{s}_t, \mathbf{a}_t \sim \pi}[\sum_t \gamma^t r(\mathbf{s}_t, \mathbf{a}_t)]$.

We further define the meta-reinforcement learning problem. Meta-training uses a distribution over MDPs, $\rho(\mathcal{T})$, from which tasks are sampled. Given a specific task $\mathcal{T}$, the agent is allowed to collect a small amount of data $\mathcal{D}_{adapt}^{(\mathcal{T})}$, and adapt the policy to obtain $\pi_\mathcal{T}$. The objective of meta-training is to maximize the expected return of the adapted policy $\mathbb{E}_{\mathcal{T} \sim \rho(\mathcal{T}), \mathbf{s}_t, \mathbf{a}_t \sim \pi_\mathcal{T}}[\sum_t \gamma^t r(\mathbf{s}_t, \mathbf{a}_t)]$.

MIER also makes use of a learned dynamics and reward model, similar to model-based reinforcement learning. In model-based reinforcement learning, a model $\hat{p}(\mathbf{s}', \mathbf{r}|\mathbf{s}, \mathbf{a})$ that predicts the reward and next state from current state and action is trained using supervised learning. The model may then be used to generate data to train a policy, using an objective similar to the RL objective above: $\arg\max_\pi \mathbb{E}_{\mathbf{s}_t, \mathbf{a}_t \sim \pi, \hat{p}}[\sum_t \gamma^t r(\mathbf{s}_t, \mathbf{a}_t)]$. Note that the expectation is now taken with respect to the policy and learned model, rather than the policy and the true MDP transition function $p(\mathbf{s}'|\mathbf{s}, \mathbf{a})$.

In order to apply model-based RL methods in meta-RL, we need to solve a supervised meta-learning problem. We briefly introduce the setup of supervised meta-learning and the model agnostic meta-learning approach, which is an important foundation of our work. In supervised meta-learning, we also have a distribution of tasks $\rho(\mathcal{T})$ similar to the meta-RL setup, except that the task $\mathcal{T}$ is now a pair of input and output random variables $(X_\mathcal{T}, Y_\mathcal{T})$. Given a small dataset $\mathcal{D}_{adapt}^{(\mathcal{T})}$ sampled from a specific task $\mathcal{T}$, the objective is to build a model that performs well on the evaluation data $\mathcal{D}_{eval}^{(\mathcal{T})}$ sampled from the same task. If we denote our model as $f(X; \theta)$, the adaptation process as $\mathcal{A}(\theta, \mathcal{D}_{adapt}^{(\mathcal{T})})$ and our loss function as $\mathcal{L}$, the objective can be written as:

$$\min_{f, \mathcal{A}} \mathbb{E}_{\mathcal{T} \sim \rho(\mathcal{T})} \left[ \mathcal{L} \left( f \left( X_\mathcal{T}; \mathcal{A} \left( \theta, \mathcal{D}_{adapt}^{(\mathcal{T})} \right) \right), Y_\mathcal{T} \right) \right]$$

Model agnostic meta-learning (Finn et al., 2017a) is an approach to solve the supervised meta-learning problem. Specifically, the model $f(X; \theta)$ is represented as a neural network, and the adaptation process is represented as few steps of gradient descent. For simplicity of notation, we only write out one step of gradient descent:

$$\mathcal{A}_{\text{MAML}} \left( \theta, \mathcal{D}_{adapt}^{(\mathcal{T})} \right) = \theta - \alpha \nabla_\theta \mathbb{E}_{X, Y \sim \mathcal{D}_{adapt}^{(\mathcal{T})}} \left[ \mathcal{L} \left( f \left( X; \theta \right), Y \right) \right]$$

The training process of MAML can be summarized as optimizing the loss of the model after few steps of gradient descent on data from the new task. Note that because $\mathcal{A}_{\text{MAML}}$ is the standard gradient descent operator, our model is guaranteed to improve under suitable smoothness conditions regardless of the task distribution $\rho(\mathcal{T})$, though adaptation to in-distribution tasks is likely to be substantially more efficient.

## 3 META TRAINING WITH MODEL IDENTIFICATION

As discussed in Section 1, MIER is built on top of two concepts, which we call **model identification** and **experience relabeling**. We first discuss how we can reformulate the meta-RL problem into a model identification problem, where we train a fast-adapting model to rapidly identify the transition dynamics and reward function for a new task. We parameterize the model with a latent context variable, which is meta-trained to encapsulate all of the task-specific information acquired during adaptation. We then train a universal policy that, conditioned on this context variable, can solve all of the meta-training tasks. Training this policy is a standard RL problem instead of a meta-RL problem, so any off-the-shelf off-policy algorithms can be used. The resulting method can immediately be used to adapt to new in-distribution tasks, simply by adapting the model's context via gradient descent, and conditioning the universal policy on this context. We illustrate the model identification part of our algorithm in the left part of Figure 1 and provide pseudo-code for our meta-training procedure in Algorithm 1.

In a meta-RL problem, where tasks are sampled from a distribution of MDPs, the only varying factors are the dynamics $p(\mathbf{s}'|\mathbf{s}, \mathbf{a})$ and the reward function $r$. Therefore, a sufficient condition for identifying the task is to learn the transition dynamics and the reward function, and this is exactly what model-based RL methods do. Hence, we can naturally formulate the meta-task identification problem as a model-based RL problem and solve it with supervised meta-learning methods.

Specifically, we choose the MAML method for its simplicity and consistency. Unlike the standard supervised MAML formulation, we condition our model on a latent context vector, and we only change the context vector when adapting to new tasks. This formulation is previously known as bias transformation for MAML (Finn et al., 2017b; Finn & Levine, 2018; Zintgraf et al., 2018). Since all task-specific information is thus encapsulated in the context vector, conditioning the policy on this context should provide it with sufficient information to solve the task. This architecture is illustrated in the left part of Figure 1. We denote the model as $\hat{p}(\mathbf{s}', \mathbf{r}|\mathbf{s}, \mathbf{a}; \theta, \phi)$, where $\theta$ is the neural network parameters and $\phi$ is the latent context vector that is passed in as input to the network.

One step of gradient adaptation can be written as follows:

$$\phi_{\mathcal{T}} = \mathcal{A}_{\text{MAML}}\left(\theta, \phi, \mathcal{D}_{adapt}^{(\mathcal{T})}\right) = \phi - \alpha\nabla_{\phi}\mathbb{E}_{(\mathbf{s},\mathbf{a},\mathbf{s}',\mathbf{r})\sim\mathcal{D}_{adapt}^{(\mathcal{T})}}[-\log\hat{p}\left(\mathbf{s}', \mathbf{r}|\mathbf{s}, \mathbf{a}; \theta, \phi\right)]. \quad (1)$$

We use the log likelihood as our objective for the probabilistic model. We then evaluate the model using the adapted context vector $\phi_{\mathcal{T}}$, and minimize its loss on the evaluation dataset to learn the model. Specifically, we minimize the model meta-loss function $J_{\hat{p}}(\theta, \phi, \mathcal{D}_{adapt}^{(\mathcal{T})}, \mathcal{D}_{eval}^{(\mathcal{T})})$ to obtain the optimal parameter $\theta$ and context vector initialization $\phi$:

$$\arg\min_{\theta,\phi} J_{\hat{p}}\left(\theta, \phi, \mathcal{D}_{adapt}^{(\mathcal{T})}, \mathcal{D}_{eval}^{(\mathcal{T})}\right) = \arg\min_{\theta,\phi}\mathbb{E}_{(\mathbf{s},\mathbf{a},\mathbf{s}',\mathbf{r})\sim\mathcal{D}_{eval}^{(\mathcal{T})}}[-\log\hat{p}\left(\mathbf{s}', \mathbf{r}|\mathbf{s}, \mathbf{a}; \theta, \phi_{\mathcal{T}}\right)]$$

The main difference between our method and previously proposed meta-RL methods that also use context variables (Rakelly et al., 2019; Duan et al., 2016) is that we use gradient descent to adapt the context. Adaptation will be much faster for tasks that are in-distribution, since the meta-training process explicitly optimizes for this objective, but the model will still adapt to out-of-distribution tasks given enough samples and gradient steps, since the adaptation process corresponds to a well-defined and convergent learning process. However, for out-of-distribution tasks, the adapted context could be out-of-distribution for the policy. We address this problem in Section 4.

Given the latent context variable from the adapted model $\phi_{\mathcal{T}}$, the meta-RL problem can be effectively reduced to a standard RL problem, as the task specific information has been encoded in the context variable. We can therefore apply any standard model-free RL algorithm to obtain a policy, as long as we condition the policy on the latent context variable.

In our implementation, we utilize the soft actor-critic (SAC) algorithm (Haarnoja et al., 2018), though any efficient model-free RL method could be used. We briefly introduce the policy optimization process for a general actor-critic method. Let us parameterize our policy $\pi_{\psi}$ by a parameter vector $\psi$. Actor-critic methods maintain an estimate of the Q values for the current policy, $Q^{\pi_{\psi}}(\mathbf{s}, \mathbf{a}, \phi_{\mathcal{T}}) = \mathbb{E}_{\mathbf{s}_t,\mathbf{a}_t\sim\pi_{\psi}}[\sum_t\gamma^t r(\mathbf{s}_t, \mathbf{a}_t)|\mathbf{s}_0 = \mathbf{s}, \mathbf{a}_0 = \mathbf{a}, \mathcal{T}]$, via Bellman backups, and improve the policy by maximizing the expected Q values under the policy, averaged over the dataset $\mathcal{D}$. The policy loss can be written as:

---

**Algorithm 1** Model Identification Meta-Training

**Input:** task distribution $\rho(\mathcal{T})$, training steps $N$, learning rate $\alpha$
**Output:** policy parameter $\psi$, model parameter $\theta$, model context $\phi$

Randomly initialize $\psi, \theta, \phi$
Initialize multitask replay buffer $\mathcal{R}(\mathcal{T}) \leftarrow \emptyset$
**while** $\theta, \phi, \psi$ *not converged* **do**
    Sample task $\mathcal{T} \sim \rho(\mathcal{T})$
    Collect $\mathcal{D}_{adapt}^{(\mathcal{T})}$ using $\pi_{\psi}$ and $\phi$
    Compute $\phi_{\mathcal{T}} = \mathcal{A}_{\text{MAML}}(\theta, \phi, \mathcal{D}_{adapt}^{(\mathcal{T})})$
    Collect $\mathcal{D}_{eval}^{(\mathcal{T})}$ using $\pi$ and $\phi_{\mathcal{T}}$
    $\mathcal{R}(\mathcal{T}) \leftarrow \mathcal{R}(\mathcal{T}) \cup \mathcal{D}_{adapt}^{(\mathcal{T})} \cup \mathcal{D}_{eval}^{(\mathcal{T})}$
    **for** $i = 1$ *to* $N$ **do**
        Sample task $\mathcal{T} \sim \mathcal{R}$
        Sample $\mathcal{D}_{adapt}^{(\mathcal{T})}, \mathcal{D}_{eval}^{(\mathcal{T})} \sim \mathcal{R}(\mathcal{T})$
        Update $\theta \leftarrow \theta - \alpha\nabla_{\theta}J_{\hat{p}}(\theta, \phi, \mathcal{D}_{adapt}^{(\mathcal{T})}, \mathcal{D}_{eval}^{(\mathcal{T})})$
        Update $\psi \leftarrow \psi - \alpha\nabla_{\psi}J_{\pi}(\psi, \mathcal{D}_{eval}^{(\mathcal{T})}, \phi_{\mathcal{T}})$
    **end**
**end**

---

$$J_{\pi}(\psi, \mathcal{D}, \phi_{\mathcal{T}}) = -\mathbb{E}_{\mathbf{s}\sim\mathcal{D},\mathbf{a}\sim\pi}[Q^{\pi_{\psi}}(\mathbf{s}, \mathbf{a}, \phi_{\mathcal{T}})]$$

Note that we condition our value function $Q^{\pi_{\psi}}(\mathbf{s}, \mathbf{a}, \phi_{\mathcal{T}})$ and policy $\pi_{\psi}(\mathbf{a}|\mathbf{s}, \phi_{\mathcal{T}})$ on the adapted task specific context vector $\phi_{\mathcal{T}}$, so that the policy and value functions are aware of which task is being performed (Schaul et al., 2015; Kaelbling, 1993). Aside from incorporating the context $\phi_{\mathcal{T}}$, the actual RL algorithm is unchanged, and the main modification is the concurrent meta-training of the model to produce $\phi_{\mathcal{T}}$.

## 4 ADAPTING TO OUT-OF-DISTRIBUTION TASKS VIA EXPERIENCE RELABELING

At meta-test time, when our method must adapt to a new unseen task $\mathcal{T}$, it first samples a small batch of data and obtain the latent context $\phi_{\mathcal{T}}$ by running the gradient descent adaptation process on the context variable, using the model identification process introduced in the previous section. While our model identification method is already a complete meta-RL algorithm, it has no guarantees of consistency. That is, it might not be able to adapt to out-of-distribution tasks, even with large amounts of data: although the gradient descent adaptation process for the model is consistent and will continue

to improve, the context variable $\phi_{\mathcal{T}}$ produced by this adaptation may still be out-of-distribution for the policy when adapting to an out-of-distribution task. However, with an improved model, we can continue to train the policy with standard off-policy RL, by generating synthetic data using the model. In practice we adapt the model for as many gradient steps as necessary, and then use this model to generate synthetic transitions using states from all previously seen meta-training tasks, with new successor states and rewards. We call this process experience relabeling. Since the model is adapted via gradient descent, it is guaranteed to eventually converge to a local optimum for any new task, even a task that is outside the meta-training distribution. We illustrate the experience relabeling process in the right part of Figure 1, and provide pseudo-code in Algorithm 2.

When using data generated from a learned model to train a policy, the model's predicted trajectory often diverges from the real dynamics after a large number of time steps, due to accumulated error (Janner et al., 2019). We can mitigate this issue in the meta-RL case by leveraging all of the data from other tasks that was available during meta-training. Although new task is previously unseen, the other training tasks share the same state space and action space, and so we can leverage the large set of diverse transitions collected from these tasks. Using the adapted model and policy, we can *relabel* these transitions, denoted $(\mathbf{s}, \mathbf{a}, \mathbf{s}', \mathbf{r})$, by sampling new actions with our adapted policy, and by sampling next states and rewards from the adapted model. The relabeling process can be written as:

---

**Algorithm 2** Experience Relabeling Adaptation

**Input:** test task $\hat{\mathcal{T}}$, multitask replay buffer $\mathcal{R}(\mathcal{T})$, Adaptation steps for context $N_{fast}$, Training steps for policy $N_p$, Training steps for model $N_m$
**Output:** policy parameter $\psi$

Collect $\mathcal{D}_{adapt}^{(\hat{\mathcal{T}})}$ from $\hat{\mathcal{T}}$ using $\pi_\psi$ and $\phi$
**for** $i = 1$ *to* $N_{fast}$ **do**
  | Update $\phi_{\mathcal{T}}$ according to Eq. 1
**end**
**while** $\psi$ *not converged* **do**
  **for** $i = 1$ *to* $N_p$ **do**
    Sample $\mathcal{T} \sim \mathcal{R}$ and $\mathcal{D}^{(\mathcal{T})} \sim \mathcal{R}(\mathcal{T})$
    $\hat{\mathcal{D}}^{(\mathcal{T})} \leftarrow \mathbf{Relabel}(\mathcal{D}^{(\mathcal{T})}, \theta, \phi_{\hat{\mathcal{T}}})$
    Train policy $\psi \leftarrow \psi - \alpha \nabla_\psi J_\pi(\psi, \hat{\mathcal{D}}^{(\mathcal{T})}, \phi_{\mathcal{T}})$
  **end**
**end**

---

$$\mathbf{Relabel}(\mathcal{D}, \theta, \phi_{\mathcal{T}}) = \{(\mathbf{s}, \mathbf{a}, \mathbf{s}', \mathbf{r}) | \mathbf{s} \in \mathcal{D}; \mathbf{a} \sim \pi(\mathbf{a}|\mathbf{s}, \phi_{\mathcal{T}}), (\mathbf{s}', \mathbf{r}) \sim \hat{p}(\mathbf{s}', \mathbf{r}|\mathbf{s}, \mathbf{a}; \theta, \phi_{\mathcal{T}})\}.$$

We use these relabeled transitions to continue training the policy. The whole adaptation process is summarized in Algorithm 2. Since the learned model is only used to predict one time step into the future, our approach does not suffer from compounding model errors. We also note that our experience relabeling method is a general tool for adapting to out-of-distribution tasks, and could be used independently of our model identification algorithm. For example, we could apply a standard, non-context-based dynamics and reward model to generate synthetic experience to finetune a policy obtained from any source, including other meta-RL methods.

Similar to our method, the MQL algorithm (Fakoor et al., 2020) also reuses data collected during meta-training time to continue improve the policy during adaptation. However, the way it reuses data is different. Given a new adaptation dataset $\mathcal{D}_{adapt}$ and replay buffer $\mathcal{R}$ containing data from other tasks, MQL estimates a density ratio $\frac{q(\mathbf{s}, \mathbf{a}, \mathbf{s}', \mathbf{r})}{q'(\mathbf{s}, \mathbf{a}, \mathbf{s}', \mathbf{r})}$, where $q$ and $q'$ are the corresponding probability density on $\mathcal{D}_{adapt}$ and $\mathcal{R}$. MQL then **re-weight** the transitions in the replay buffer using this density ratio to compute the loss for the policy and value function. This implicitly assumes that the data distributions of different tasks **share the same support**, i.e. $q(\mathbf{s}, \mathbf{a}, \mathbf{s}', \mathbf{r}) > 0$ for transitions sampled from other tasks in the replay buffer. This assumption may not be true in many practical domains. We will show empirically that indeed when this assumption is violated, MQL is not able to adapt effectively. Our method avoids this problem by using an adapted model to **relabel** the data. By generating a synthetic next state and reward, the relabeled transition would be useful for the new task, even if the original transition is not. The only assumption for our method is that different tasks **share the same state and action space**, which is true for most meta-RL domains.

## 5 RELATED WORK

Meta-reinforcement learning algorithms extend the framework of meta-learning (Schmidhuber, 1987; Thrun & Pratt, 1998; Naik & Mammone, 1992; Bengio et al., 1991) to the reinforcement learning setting. Model-free encoder-based methods encode the transitions seen during adaptation into a latent context variable, and the policy is conditioned on this context to adapt it to the new task.

The context encoding process is often done via a recurrent network (Duan et al., 2016; Wang et al., 2016; Fakoor et al., 2020; Stadie et al., 2018), an attention mechanism (Mishra et al., 2017), or via amortized variational inference (Rakelly et al., 2019; Humplik et al., 2019). While inference is efficient for handling in-distribution tasks (Fig. 2), it is not effective for adaptation to out-of-distribution tasks (Fig. 4). On the other hand, MIER can handle out-of-distribution tasks through the use of a consistent gradient descent learner for the model, followed by a consistent (non-meta-trained) off-policy reinforcement learning method.

Model-free gradient-based meta-RL methods (Finn et al., 2017a; Rothfuss et al., 2018; Rusu et al., 2018; Liu et al., 2019; Gupta et al., 2018; Sung et al., 2017; Houthooft et al., 2018) implement gradient descent as the adaptation process. However, they are based on on-policy RL algorithms, and thus require a large number of samples for training and adaptation (Fig. 2). There are also works that combine gradient-based and encoder-based methods (Lan et al., 2019). However, such methods still suffer from the same sample efficiency problem as other gradient based methods, because of the use of on-policy policy gradients. Our method avoids this problem by combining a gradient-based supervised meta-learning algorithm with an off-policy RL algorithm to achieve sample efficiency comparable to that of off-policy encoder-based methods.

There has been some work that uses off-policy policy gradients for sample efficient meta-training (Mendonca et al., 2019), but this still requires quite a few trajectories for policy gradient based adaptation at test time. MIER avoids this by reusing the experiences collected during training to enable fast adaptation with minimal amount of additional data.

Model based meta-RL methods meta-train a model rather than a policy (Nagabandi et al., 2018; Sæmundsson et al., 2018). At test time, when the model is adapted to a particular task, standard planning techniques such as model predictive control (Williams et al., 2015; Camacho & Alba, 2013) are often applied to select actions. Unfortunately, pure model-based meta-RL methods typically attain lower returns than their model-free counter-parts, particularly for long-horizon tasks. Our method can attain comparatively higher final returns because we only use one-step predictions from our model to provide synthetic data for a model-free RL method (Fig. 4). This resembles methods that combine model learning with model-free RL in single-tasks settings (Sutton, 1991; Janner et al., 2019).

## 6  EXPERIMENTAL EVALUATION

We aim to answer the following questions in our experiments: **(1)** Can MIER meta-train efficiently on standard meta-RL benchmarks, with meta-training sample efficiency that is competitive with state-of-the-art methods? **(2)** How does MIER compare to prior meta-learning approaches for extrapolation to meta-test tasks with out-of-distribution (a) reward functions and (b) dynamics? **(3)** How important is experience relabeling in leveraging the model to train effective policies for out-of-distribution tasks?

To answer these questions, we first compare the meta-training sample efficiency of MIER to existing methods on several standard meta-RL benchmarks. We then test MIER on a set of out-of-distribution meta-test tasks to analyze extrapolation performance. We also compare against a version of our method without experience relabeling, in order to study the importance of this component for adaptation. All experiments are run with OpenAI gym (Brockman et al., 2017) and use the mujoco simulator (Todorov et al., 2012). Additional implementation and experiment details including hyperparameters are included in Appendix C.

### 6.1  META-TRAINING SAMPLE EFFICIENCY ON META-RL BENCHMARKS

We first evaluate MIER on standard meta-RL benchmarks, which were used in prior work (Finn et al., 2017a; Rakelly et al., 2019; Fakoor et al., 2020) and show the result in Figure 2. We compare to PEARL (Rakelly et al., 2019), which uses an off-policy encoder-based method, but without consistent adaptation, meta Q-learning (MQL) (Fakoor et al., 2020), which also uses an encoder, MAML (Finn et al., 2017a) and PRoMP (Rothfuss et al., 2018), which use MAML-based adaptation with on-policy policy gradients, and RL2 (Duan et al., 2016), which uses an on-policy algorithm with an encoder. We plot the meta-test performance after adaptation (on **in-distribution** tasks) against the number of meta-training samples, averaged across 3 random seeds. On these standard tasks, we run a variant of our full method which we call MIER-wR (MIER without experience relabeling). Note that our implementation generates two hundred thousand exploration samples from the initial policy during

Figure 2: Performance on standard meta-RL benchmarks. Return is evaluated over the course of the *meta-training* process on meta-test tasks that are **in-distribution**.

meta-training prior beginning meta-training, following the same convention as PEARL (Rakelly et al., 2019), whereas MQL does not generate these samples, which explains the difference in terms of where training begins (most visible in Half-Cheetah-Vel), resulting in a constant offset of the learning curve along the x-axis. Our method achieves performance comparable to or better than the best prior methods, indicating that our model identification method provides a viable meta-learning strategy that compares favorably to state-of-the-art methods. However, the primary focus of our paper is on adaptation to **out-of-distribution** tasks, which we analyze next.

## 6.2 ADAPTATION TO OUT-OF-DISTRIBUTION TASKS

Next, we evaluate how well MIER can adapt to out-of-distribution, both on tasks with varying reward functions and tasks with varying dynamics. We compare the performance of our full method (MIER), and MIER without experience relabeling (MIER-wR), to prior meta-learning methods for adaptation to out-of-distribution tasks. All algorithms are meta-trained with the same number of samples (2.5M for Ant Negated Joints, and 1.5M for all other domains) before evaluation. For performance of algorithms as a function of data used for meta-training, see Figure 6 in Appendix B.

**Extrapolation over reward functions:** To evaluate extrapolation to out-of-distribution rewards, we first test on the velocity extrapolation environments of HalfCheetah introduced by Fakoor et al. (2020). Half-Cheetah-Vel-Medium meta-trains on tasks where the cheetah is required to run at target speeds ranging from 0 to 2.5 m/s, while Half-Cheetah-Hard meta-trains on speeds from 0 to 1.5 m/s, as depicted in Figure 3(a). In both settings, the meta-test set has target speeds sampled from 2.5 to 3 m/s. In Figure 4, we see that our method matches MQL on the easier Half-Cheetah-Vel-Medium environment, and outperforms all prior methods including MQL on the Half-Cheetah-Vel-Hard setting, where the meta-test tasks are further outside the distribution of meta-training tasks. We see that experience relabeling improves performance in both settings.

We also evaluate reward function extrapolation on the Ant tasks, where meta-training task directions are sampled from 3 quarters of the circle, and the meta-test set contains the remaining quadrant, as illustrated in Figure 3(b). We see in Figure 4 that our method outperforms PEARL and MAML by a large margin in this setting, while MQL attains better performance. We provide a more fine-grained analysis of adaptation performance on different tasks in the test set in Figure 6. We see that while the performance of all methods degrades as validation tasks get farther away from the training distribution, MIER and MIER-wR perform consistently better than MAML and PEARL. Note, however, that the reward extrapolation represents in some sense the easiest setting for extrapolate, since the dynamics are preserved across all tasks, and our method's performance would likely improve with better models. In the next paragraph, we study the more challenging setting, where extrapolation requires adapting to different dynamics.

**Extrapolation over dynamics:** To study adaptation to out-of-distribution dynamics, we constructed variants of the HalfCheetah and Ant environments where we negate the control of randomly selected groups of joints as shown in Figures 3(c) and 3(d). During meta-training, we never negate the last joint so we can construct out-of-distribution tasks by negating this last joint together with a randomly chosen subset of the others. For HalfCheetah, we negate 3 joints at a time from among the first 5 during meta-training, and always negate the 6th joint (together with a random subset of 2 of the other 5) for testing, such that there are 10 meta-training tasks and 10 out-of-distribution evaluation tasks. For Ant, we negate 4 joints from among the first 7 during meta-training, and always negate the 8th (together with a random subset of 3 of the other 7) for evaluation, resulting in 35 meta-training tasks and 35 evaluation tasks, out of which we randomly select 15.

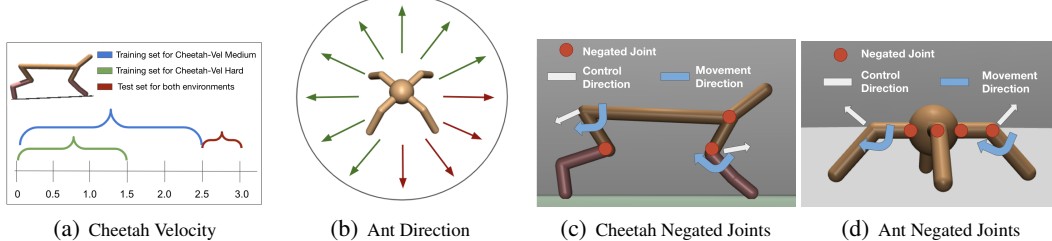

Figure 3: Illustration of out-of-distribution adaptation tasks: (a) Cheetah-Velocity Medium (target velocity training set in blue, test set in red) and Cheetah-Velocity Hard (target velocity training set in green, test set in red), (b) Ant Direction (target direction training tasks in green, test tasks in red), (c) Cheetah Negated Joints and (d) Ant Negated Joints. Training and test sets are indicated in the figure for (a) and (b). In the negated joint environments, the control is negated to a set of randomly selected joints, and the movement direction when control is applied is depicted for both negated and normal joints.

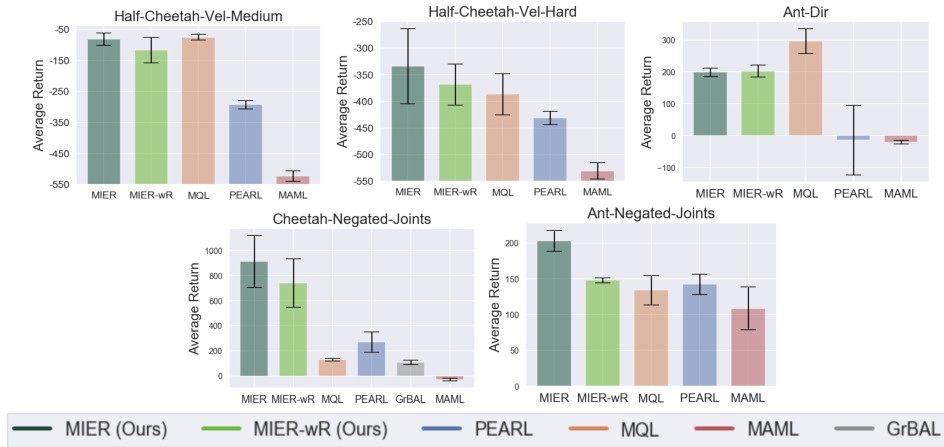

Figure 4: Performance on out-of-distribution tasks. All algorithms are meta-trained with the same amount of data, and then evaluated on out-of-distribution tasks. Cheetah-Velocity and Ant-Direction environments have varying reward functions, while Cheetah-Negated-Joints and Ant-Negated-Joints have different dynamics.

In addition to MQL, PEARL and MAML, we compare against GrBAL (Nagabandi et al., 2018), a model based meta-RL method. We could not evaluate GrBAL on the reward extrapolation tasks, since it requires the analytic reward function to be known, but we can compare to this method under varying dynamics. From Figure 4, we see that performance on Cheetah-Negated-Joints with just context adaptation (MIER-wR) is substantially better than prior methods, and there is further improvement by using the model for relabeling (MIER). On the more challenging Ant-Negated-Joints environment, MIER-wR shows similar performance to MQL and PEARL, and leveraging the model for relabeling again leads to substantially better performance for MIER. We note that in this harder dynamics extrapolation setting, the full MIER method attains a significant improvement over MQL, which we hypothesize is based the shared support assumption described in Section 4 is violated in these two environments.

## 7 CONCLUSION

In this paper, we introduce a consistent and sample efficient meta-RL algorithm by reformulating the meta-RL problem as **model identification**, and then described a method for adaptation to out-of-distribution tasks based on **experience relabeling**. Our algorithm can adapt *consistently* to out-of-distribution tasks by adapting the model first, relabeling all data from the meta-training tasks with this model, and then fine-tuning on that data using a standard off-policy RL method. While model identification and experience relabeling can be used independently, with the former providing for a simple meta-RL framework and the latter providing for adaptation to out-of-distribution tasks, we show that combining these components leads to good results across a range of challenging meta-RL problems that require extrapolating to out-of-distribution tasks at meta-test time.

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

# Appendices

## A   THE DIFFICULTY OF COMBINING GRADIENT-BASED META-LEARNING WITH VALUE-BASED RL METHODS

One straightforward idea of building a sample efficient off-policy meta-RL algorithm that adapts well to out-of-distribution task is to simply combine MAML with an off-policy actor-critic RL algorithm. However, this seemingly simple idea is very difficult in practice, mainly because of the difference between Bellman backup iteration used in actor-critic methods and gradient descent. Consider the Bellman backup of Q function $Q^\pi$ for policy $\pi$,

$$Q^\pi(\mathbf{s}, \mathbf{a}) \leftarrow \mathbf{r}(\mathbf{s}, \mathbf{a}) + \gamma \mathbb{E}_{\mathbf{s}' \sim p(\mathbf{s}'|\mathbf{s}, \mathbf{a}), \mathbf{a}' \sim \pi(\mathbf{a}'|\mathbf{s}')}[Q^\pi(\mathbf{s}', \mathbf{a}')]$$

which backs up the next state Q value to the current state Q value. One iteration of Bellman backup can only propagate value information backward in time for one timestep. Therefore, given a trajectory with horizon $T$, even if we can perform the backup operation exactly at every iteration, at least $T$ iterations of Bellman backup is required for the Q function to converge. Therefore, it cannot be used as the inner loop objective for MAML, where only a few steps of gradient descent is allowed. In practice $T$ is usually 200 for MuJoCo based meta-RL domains, and applying MAML with 200 steps of inner loop is certainly intractable. If we only perform $K$ steps of Bellman backup for the inner loop, where $K$ is a small number, we would obtain a Q function that is greedy in $K$ steps, which gives us very limited performance. In fact, we realized this limitation only after implementing this method, where we were never able to get it to work in even the easiest domain.

## B   SAMPLE EFFICIENCY AND ANALYSIS FOR EXTRAPOLATION TASKS

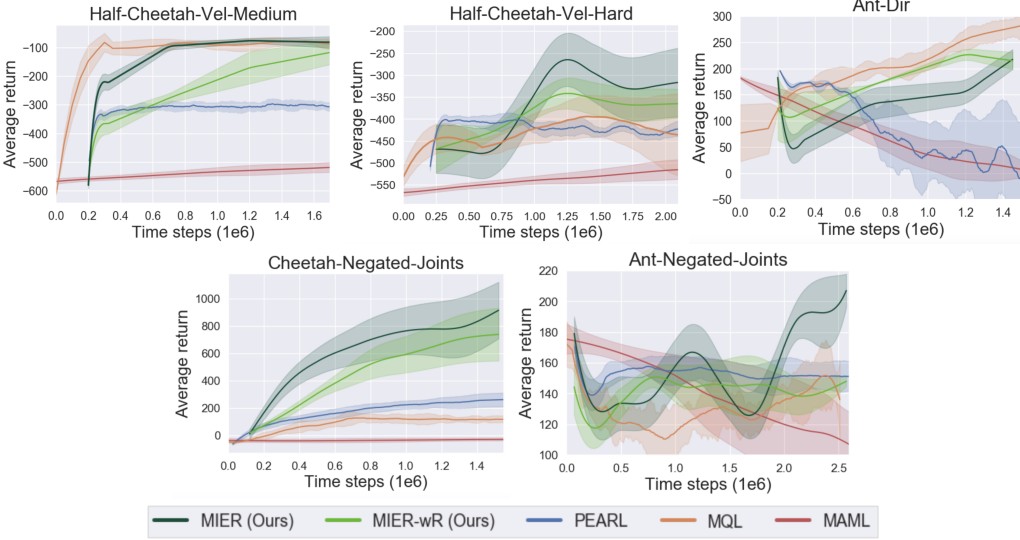

Figure 5: Extrapolation performance on OOD tasks. In all experiements, we see our method exceeds or matches the performance of previous state-of-the-art methods. We also observe that experience relabeling is crucial to getting good performance on out-of-distribution tasks.

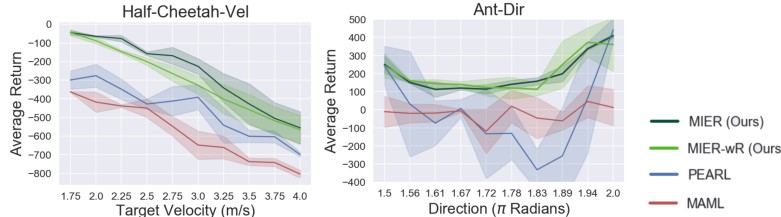

Figure 6: Performance evaluated on validation tasks of varying difficulty. For Cheetah Velocity, the training distribution consists of target speeds from 0 to 1.5 m/s, and so tasks become harder left to right along the x axis. Ant Direction consists of training tasks ranging from 0 to 1.5 $\pi$ radians, so the hardest tasks are in the middle.

## C  IMPLEMENTATION DETAILS

Please see the released codebase for code to meta-train models and extrapolate to out-of-distribution tasks. We also include code for the simulation environments included in the paper.

### C.1  DATASETS

All experiments are run with OpenAI gym (Brockman et al., 2017), use the mujoco simulator (Todorov et al., 2012) and are run with 3 seeds (We meta-train 3 models, and run extrapolation for each). The metric used to evaluate performance is the average return (sum of rewards) over a test rollout. The horizon for all environments is 200. For the meta-RL benchmarks (Fig. 2), performance on test tasks is plotted versus number of samples meta-trained on. The out-of-distribution plots (Fig. 4 and 6) report performance of all algorithms meta-trained with the same number of samples (2.5M for Ant Negated Joints, and 1.5M for all other domains). For the standard meta-RL benchmark tasks, we use the settings from PEARL (Rakelly et al., 2019) for number of meta-train tasks, meta-test tasks and data points for adaptation on a new test task. For the out-of-distribution experiments, the values used for datasets are listed in Table 1. The description of the meta-train and meta-test task sets for out-of-distribution tasks is included in Section 6.2.

### C.2  EXTRAPOLATION EXPERIMENT DETAILS

For the settings with varying reward functions, the state dynamics does not differ across tasks, and so we only meta-train a reward prediction model. We only relabel rewards and preserve the (state, action, next state) information from cross task data while relabelling experience in this setting. For domains with varying dynamics, we meta-learn both reward and state models.

When continually adapting the model to out of distribution tasks, we first take a number of gradient steps (**N**) that only affect the context , followed by another number of gradient steps (**M**) that affect all model parameters. We also note that if the model adaptation process overfits to the adaptation data, using generated synthetic data will lead to worse performance for the policy. To avoid this, we only use 80% of the adaptation data to learn the model, and use the rest for validation. The model is used to produce synthetic data for a task only if the total model loss on the validation set is below a threshold (set to -3).

Table 1: Settings for out-of-distribution environments

| Environment | Meta-train tasks | Meta-test tasks | Data points for adaptation | N | M |
|---|---|---|---|---|---|
| Cheetah-vel-medium | 100 | 30 | 200 | 10 | 100 |
| Cheetah-vel-hard | 100 | 30 | 200 | 10 | 100 |
| Ant-direction | 100 | 10 | 400 | 20 | 0 |
| Cheetah-negated-joints | 10 | 10 | 400 | 10 | 0 |
| Ant-negated-joints | 10 | 10 | 400 | 10 | 0 |
| Walker-rand-params | 40 | 20 | 400 | 10 | 100 |

## C.3 HYPER-PARAMETERS

For the MIER experiments hyper-parameters are kept mostly fixed across all experiments, with the model-related hyperparameters set to default values used in the Model Based Policy Optimization codebase (Janner et al., 2019), and the policy-related hyperparameters set to default settings in PEARL (Rakelly et al., 2019), and their values are detailed in Table 2. We also ran sweeps on some hyper-parameters, detailed in Table 3.

For the baselines, we used publicly released logs for the benchmark results, and ran code released by the authors for the out-of-distribution tasks. Hyper-parameters were set to the default values in the codebases. We also swept on number of policy optimization steps and context vector dimension for PEARL, similar to the sweep in Table 3.

### Table 2: Default Hyper-parameters

(a) Model-related

| Hyperparameter | Value |
|---|---|
| Model arch | 200-200-200-200 |
| Meta batch size | 10 |
| Inner adaptation steps | 2 |
| Inner learning rate | 0.01 |
| Number of cross tasks for rela-belling | 20 |
| Batch-size for cross task sampling | 1e5 |
| Dataset train-val ratio for model adaptation | 0.8 |

(b) Policy-related

| Hyperparameter | Value |
|---|---|
| Critic arch | 300-300-300 |
| Policy arch | 300-300-300 |
| Discount factor | 0.99 |
| Learning rate | 3e-4 |
| Target update interval | 1 |
| Target update rate | 0.005 |
| Sac reward scale | 1 |
| Soft temperature | 1.0 |
| Policy training batch-size | 256 |
| Ratio of real to synthetic data for continued training | 0.05 |
| Number of policy optimization steps per synthetic batch generation | 250 |

### Table 3: Hyper-parameter sweeps

| Hyper-parameter | Value | Selected Values |
|---|---|---|
| Number of policy optimization steps per meta-training iteration | 1000, 2000, 4000 | 1000 |
| Context vector dimension | 5, 10 | 5 |
| Gradient norm clipping | 10, 100 | 10 |

All experiments used GNU parallel (Tange, 2011) for parallelization, and were run on GCP instances with NVIDIA Tesla K80 GPUS.

