# OpenReview forum: "Meta-Reinforcement Learning Robust to Distributional Shift via Model Identification and Experience Relabeling"
_ICLR.cc/2021/Conference — Reject_

### Official Review · AnonReviewer1 · 2020-10-27
**This paper proposes a convincing algorithm to adapt to both in-distribution and out-of-distribution tasks, while the expressiveness of the dynamics model limits the adaptation ability of the algorithm.**

**Rating:** 6
**Confidence:** 4

**Review:**

The authors propose a novel meta-RL algorithm that can both meta-train and adapt to new tasks in an off-policy way. Multiple ideas are incorporated into the algorithm. Meta-learning parameters for both the context variable and the policy then updating them through gradient descent endows the policy with strong learning ability, while the dynamics model allows the reuse of training data by relabeling them. The authors conduct experiments in two types of task settings, which are in-distribution tasks and out-of-distribution tasks. Experiments show that MIER outperforms some existing methods on out-of-distribution tasks in terms of sample efficiency, while achieving comparable results on in-distribution tasks. The work is original, and has a certain contribution to the meta-RL setting.

## Pros:
1. Adaptation ability by modeling the dynamics. Leveraging the strong ability of gradient based models and consistency of the dynamics model, MIER can adapt to both in-distribution and out-of-distribution tasks.
2.Sample efficiency by reusing training data. Most gradient based methods require a large number of samples to calculate the loss function for gradient descent. Although a similar off-policy idea has been adopted in MQL, the authors use the training data in another way, which is by relabeling them. This technique weakens the assumption in MQL, thus is suitable for both in-distribution and OOD tasks.
3. Clarity of the paper. The authors clearly describe the motivation of the proposed algorithm, which is to achieve sample efficiency on OOD tasks. The authors extend the meta-RL problem setting to in-distribution and out-distribution tasks with varying reward functions and dynamics, and conduct experiments on these settings with multiple existing methods. The experimental results are sufficient to support most of the claims.

## Cons:
1. Assumption of simple MDP. The authors assume that a few steps of exploration are enough to capture the key information of tasks and guide the dynamics model, which might be insufficient in more complex changes in the MDP.
2. Complexity of the algorithm. In the experiments, the authors treat the information whether the task is in-distribution or out-of-distribution as known, which is unrealistic in most cases. The full adaptation process, which requires two stages of policy gradients and one stage of data relabeling, is much more complex than that of most existing methods.
3. Limitation of the dynamics model on OOD tasks. The adaptation to OOD tasks is based on the assumption that the dynamics model gives true or near-true transitions, which limits MIER’s ability. In the experiments, the authors conduct experiments that do not change some properties of MDP, more specifically, the state transition remains the same given the desired action. For some other meta-tasks where the fundamental of MDP changes, e.g. Walker-Rand-Params, the dynamics model may output undesired state-action pairs on the unseen tasks since it was not trained on these MDPs.
4. Uncertainty of MIER. Fig. 5 shows a strong uncertainty of the training curves and the training process does not seem to have converged for all experiments but Half-Cheetah Velocity. Since the paper focuses on adaptation to OOD tasks, the authors should discuss this phenomenon in the main body of the paper.
5. Necessity of data relabeling. Fig. 6 raises a question for the necessity of data relabeling in terms of the final performance, since both MIER and MEIR-wR converge to similar performance in most OOD cases.

## Questions:
1. Assume the information whether the task is in-distribution or out-of-distribution is unknown, so that data relabeling is performed anyway, will MIER still achieve a good performance on the in-distribution tasks?

Although the paper is well-organized and mostly well-written, I would like to point out some typos:
1. On page 5, in Algorithm 2: N_m is not mentioned in the algorithm.
2. In section “Experimental Evaluation” on page 6 & In C.1, Appendix C: “mujoco” -> “MuJoCo” .
3. In section “Citation”: The authors should pay more attention to the rules of references.

---

> ### Author Response · Authors · 2020-11-21
> **Author Response: In-distribution Tasks and Varying Dynamics Tasks**
>
>
> We thank the reviewer for the detailed comments and constructive feedback. We first make a brief summary of our response, and then address the detailed concerns that the reviewer raises below.
>
> The reviewer raises questions about the effectiveness of our method on in-distribution test tasks, and the ability to extrapolate to new tasks with varying dynamics. We first want to point out that the experience relabeling can be applied agnostically to both in-distribution and out-of-distribution tasks. For varying dynamics tasks, the Ant-Negated-Joint and Cheetah-Negated-Joint environments all have varying transition dynamics. The empirical results indicate that with experience relabeling, our method can significantly outperform prior methods for out-of-distribution tasks.
>
>
> **"The authors assume that a few steps of exploration are enough to capture the key information of tasks and guide the dynamics model, which might be insufficient in more complex changes in the MDP."**
>
> We agree with the reviewer that in certain environments the information might not be sufficient given only a few trajectories. The environments we evaluated our algorithm on follow the conventions of prior meta-RL research and do not have this difficulty. In these environments we show that our method extrapolates well to out-of-distribution tasks better than prior methods. The exploration of meta-RL is indeed an important problem and we would like to leave it to future research.
>
> **"Complexity of the algorithm. In the experiments, the authors treat the information whether the task is in-distribution or out-of-distribution as known, which is unrealistic in most cases."**
>
> We agree with the reviewer that the relabeling process is indeed more computationally expensive than the adaptation process of prior methods. However, we believe that this would not be a bottleneck for applying our method to real-world problems since training on most tasks is bottlenecked by data collection rather than by computation. We also want to clarify that the knowledge of in-distribution or out-of-distribution is not necessary, as one can also run the full adaptation process with relabeling for in-distribution tasks.
>
> **"Limitation of the dynamics model on OOD tasks."**
>
> We agree with the reviewer that successful adaptation of MIER relies on successful adaptation of the model. However, we do want to clarify that we are not making the assumption that “the state transition remains the same given the desired action.” Specifically, the Cheetah-Negated-Joint and Ant-Negated-Joint environment experiments all have varying transition dynamics in the MDP, and in these environments MIER significantly outperforms prior methods. Since the loss of our adaptation process considers both the reward and next state (equation 1), MIER is able to adapt to new tasks with different reward functions and transition dynamics.
>
> **"Necessity of data relabeling."**
>
> In all 4 out of 5 out-of-distribution environments, MIER with data relabeling performs better than MIER without data relabeling, especially for the environments (Cheetah-Negated-Joints and Ant-Negated-Joints) where the dynamics is changing, since it is much harder to extrapolate to out-of-distribution dynamics compared to out-of-distribution reward functions.
>
> **"Assume the information whether the task is in-distribution or out-of-distribution is unknown, so that data relabeling is performed anyway, will MIER still achieve a good performance on the in-distribution tasks?"**
>
> The knowledge of in-distribution or out-of-distribution is not necessary to apply our method, and one can use relabeling even for in-distribution tasks. For the tasks that are in distribution, the adaptation procedure will produce the right context to specify the dynamics and reward (since it is trained to do so), and hence the synthetic data produced by the model can be used for relabeling. However, this isn’t necessary, since the context contains all necessary information for the policy to identify the task. In practice we found that relabeling for in-distribution tasks produces no benefit, so we didn’t include it in the paper.

---

### Official Review · AnonReviewer3 · 2020-10-28

**Rating:** 5
**Confidence:** 3

**Review:**

Summary:
This paper presents MIER, a novel meta-RL method that is designed specifically to be able to deal with out-of-distribution test tasks. The idea is to learn two separate modules: a dynamics/reward model that consists of model parameters $\theta$ and context parameters $\phi$. This model is trained using a MAML-like gradient-based supervised meta-learning algorithm, and updates the context vector $\phi$ using gradient descent per task (and the initialisations of $\phi$, $\theta$ are meta-trained), together with a policy that is conditioned on the "context vector" $\phi$, which is task-specific after the gradient update. In order to adapt to out-of-distribution tasks at test time, the dynamics/reward model can (1) be trained with the new data, updating both $\phi$ and $\theta$, and (2) be used to train the policy using data obtained from he new model.

Impression:
The problem setting of out-of-distribution adaptation in metaRL is very interesting and relevant to the community. I think the idea of separating the model and the policy makes a lot of sense, since the model can be used to generate synthetic data on a new task for training the policy. I'm a bit underwhelmed by the empirical evaluation, and on the one hand I have a lot of open questions about what MIER is capable of doing, and on the other hand I do think that MIER *can* potentially do quite interesting things but the experiments in the paper don't really show this. I find the "extrapolation over dynamics" experiments most interesting. Here the MIER method really shines compared to other approaches. I think the paper could be made stronger by investigating the properties of MIER on a wider set of interesting experiments, and really analysing what it does and why it works well for certain problems (see concrete questions/suggestions below). I also feel like the adaptation to out-of-distribution tasks is a bit heuristic (see question below) and it's unclear to me how this would work on other / real world problems. Overall, I think this method is of interest to the community, but I also think the impact could be much higher with more discussion and empirical analysis. Hence this is a borderline paper for me at this moment.

Questions:
1. In practice, you would have to determine whether or not to update $\theta$ in addition to $\phi$, and for how many gradient steps.
 a. How do you do that, and how can you prevent under- and over-fitting?
 b. Since $\theta$ isn't trained to adapt quickly to new tasks (unlike $\phi$), could it happen that with little data this actually is a really bad initialisation and I won't be able to appropriately adapt? So even though MAML is consistent, this doesn't really help me if I don't have a lot of data.
 c. If you're looking at an out-of-distribution task, do you update both $\phi$ and $\theta$? Do you update both at the same time, or first $\phi$ and when that's not enough you update $\theta$? I guess you'd want to make sure that *most* of the new info goes into $\phi$ since that will make training the policy with the learned model easier. Is that true?
2. The pre-update policy is responsible for data collection. I expect this to be somewhat random, which works well in settings where even random rollouts contain enough information about the task. But what if that's not the case? I can see this becoming an issue in settings with sparse rewards. Do you have a sense for the exploratory behaviour that the pre-adaptation policy exhibits when collecting $\mathcal{D}_{adapt}$? In some follow-up papers to MAML (ProMP https://arxiv.org/abs/1810.06784, E-MAML https://arxiv.org/pdf/1803.01118.pdf) it has been shown that the pre-adaptation behaviour in standard MAML can be quite poor. In the environments you evaluated MIER on, the rewards are so dense that even on out-of-distribution tasks, a somewhat random pre-adaptation policy can gather data that is useful to update the model. How would you handle cases where this is no longer the case (e.g., sparse reward settings or environments that require better spatial exploration to learn about the task)?
3. In the abstract you write "Current approaches are either not able to extrapolate well, or can do so at the expense of requiring extremely large amounts of data for on-policy meta-training". Can you explain this a bit more? Why is an on-policy more likely able to extrapolate well compared to an off-policy method? Do you have examples of methods where this can be seen empirically?
4. What is the end-performance of RL2, MAML and ProMP when trained for longer? I agree that sample efficiency of MIER is a bonus, but I think we would get a better sense of how the model works and compared to baselines if we let those baselines converge.
5. I disagree with some statements about model consistency:
 - Paragraph after Eq 2: "the model will still adapt to out-of-distribution tasks given enough samples and gradient steps, since the adaptation process corresponds to a well-defined and convergent learning process."
 - First paragraph of section 4: "although the gradient descent adaptation process is consistent and will continue to improve (...)".
 - The thing I disagree about is that updating $\phi$ only is not a consistent learning algorithm, because $\phi$ is only an input vector, and might not be expressive enough such that a completely new task can eventually be learned. So to my understanding, this argument only holds for $\theta$ (+$\phi$). Could the authors comment on this?

Suggestions:
- It would be nice to add numbers to all equations, not just the first one because this makes it easier to talk about them. In my review I refer to equation numbers by just counting them.
- Sometimes I'm a little bit confused what you mean when you write "model". It's often the dynamics/reward model, but it's easily confused with the policy (e.g. last paragraph in introduction), and it's also not always clear if you refer to $\theta$ or $\phi$ (e.g., second sentence after Eq 2;  2nd+3rd sentence on page 5; and when you talk about model consistency). The readability of the paper can be improved by making this more explicit by saying "the dynamics model" or "the model parameters $\theta$" where appropriate.
- Zintgraf et al (2019) show that you don't need to meta-learn the initialisation of $\phi$, but you can always set it to to $0$ before the inner-loop update, which saves memory and is easier to implement. I can imagine that this helps with learning the policy because the values of $\phi$ might fluctuate less during meta-training. The policy that collects $\mathcal{D}_{adapt}$ then also *always* gets $\phi=0$ as input, which again could stabilise training. It might also be a way forward to look at ways to learn a good exploration policy (because now it's easier for the policy to *know* that it hasn't gathered any information yet). Might be worth a try!
- I found it weird that you compare to GrBAL on one of the environments, since it's much worse anyway. If you add this method, then it would help to explain why you compare to it, and what you can take away from this comparison. (And if there's no take-away, consider moving it to the appendix because it just distracts otherwise.)
- The experiments section has little discussion about why MIER works the way it does, in comparison to other methods. Especially on the "extrapolation over dynamics" section, where MIER really shines, a more in-depth discussion or even analysis of what is going on would really strengthen the paper, and help the read understand the method. E.g., for the "shared support assumption hypothesis" that you mention compared to MQL - can you design an algorithm where you explicitly test this?
- It would also be interesting to add an analysis / plot on how good the dynamics model is doing when it is updated on a new task, and how much data it needs to learn this.

--------------------------------------------------------------------------------------------------
UPDATE

I read the other reviews and the author's responses. Thanks for replying to my questions, that made some things clearer!

Some final comments:
- "We performed a hyper-parameter search to find the best settings of the steps to take on $\phi$ and $\theta$ for each of the environments. To ensure that we don’t over-fit the model when adapting for extrapolation, we use 80% of the available test-data to train the model, and use the rest for validation." - If I understand this correctly you tune the number of gradient steps on $\phi$/$\theta$ on part of the out-of-distribution adaptation data, and evaluate on the rest. That feels somewhat restrictive to me and might not always be possible - in which case your results are a bit optimistic. In the response to R1 you wrote that "the knowledge of in-distribution or out-of-distribution is not necessary to apply our method", but in this case the question is how to determine how to decide how often to adapt $\phi$, then $\theta$, before training the policy. At the very least, this should be discussed in more detail in the main part of the paper, given that this is central to your method.
- Thanks for pointing me to the appendix of the paper that has the end-performance of your baselines. However I do think these should be included in your paper in order for the reader to get the full picture. There might be differences in implementation (as an example, some papers use different horizon lengths in MuJoCo) so it's not always possible to compare numbers across papers, and your paper should be self-sufficient.
- Not meta-learning the initialisation of $\phi$ (but set it to zero before every inner-loop update) seems like an important detail to me - this means in Eq (2) you don't actually take the gradient w.r.t. $\phi$. It might also be an important to stabilise policy learning (because $\phi$ doesn't change too much over the course of meta-training). Even if it is an implementation detail, it should at least be mentioned in the appendix so that somebody who wants to re-implement your method can reproduce results!
- Using something like [2] for good pre-adaptation exploration is a good idea, I agree! (And I agree it's out of scope, but might be worth mentioning in the paper.)

I think overall the idea is promising but the paper falls short in terms of how the method is evaluated. I feel like too many things are buried in the appendix, and it remains unclear to me if MIER can, under realistic circumstances, really adapt to OoD tasks. I agree with R2 that Meta-World might be a suitable benchmark to evaluate MIER on, since  the training and tests tasks are distinct and there's a clear evaluation protocol for ML10 and ML45 (you have to adapt within 10 episodes) which would make it easier to compare to existing methods.

Given the above I stand by my initial rating.

---

> ### Author Response · Authors · 2020-11-21
> **Author Response: Hyperparameter Tuning, Initialization and Exploration**
>
> We thank the reviewer for the detailed comments and constructive feedback. We address the concerns the reviewer has below.
>
>
> **"Looking at an out-of-distribution task, do you update both $\theta$ and $\phi$ ?
> How do you determine whether or not to update $\theta$ in addition to $\phi$ , and for how many gradient steps ? How can you prevent under- and over-fitting?”**
>
> These are very pertinent questions raised about the details of the extrapolation process. As described in appendix C.2, we first take a number of gradient steps that only affect the context , followed by further gradient steps that affect all model parameters. We first adapt only the context because the context is meta-trained for fast adaptation. This is followed by adapting model parameters because it is possible that the learned context is not expressive enough to completely specify the out of distribution test task.
> We performed a hyper-parameter search to find the best settings of the steps to take on $\phi$ and $\theta$ for each of the environments. To ensure that we don’t over-fit the model when adapting for extrapolation, we use 80% of the available test-data to train the model, and use the rest for validation.
>
> **"Could it happen that with little data this actually is a really bad initialization and I won't be able to appropriately adapt? “**
>
> We agree that it is possible that the MAML might learn a bad initialization for the dynamics model. However, the problem is also present for any other meta-RL methods. In practice we use a small hold-out validation set to evaluate the performance of the model during adaptation, as described in Appendix C.
>
> **"Exploratory behavior of pre-adaptation policy”**
>
> We would like to point out the data collection for RL is handled by the off-policy RL method that trains the context-conditioned policy, which is decoupled from the meta-training process that is used to train the dynamics model. Hence the pre-adaptation policy’s behavior is similar to that of Pearl ([3]).
>
> It is true that as presented, our approach might struggle with dealing with sparse rewards. However, this is orthogonal to the main focus of our method: extrapolation to OOD tasks. This problem can potentially be addressed in future research by augmenting our method with exploration strategies such as [2].
>
> **"Why is an on-policy more likely able to extrapolate well compared to an off-policy method? “**
>
> Previous work ([5]) shows that gradient-based meta-learning method extrapolates to out-of-distribution tasks better than encoder-based methods, since gradient descent is a consistent learning algorithm and the learn encoder is usually not. Most of the current on-policy methods are based on MAML style gradient based adaptation, while almost all off-policy meta-RL methods are encoder-based. Hence most on-policy methods tend to extrapolate better than off-policy methods.
>
> **"End-performance of RL2, MAML, ProMP**"
>
> We would like to refer the reviewer to Fig. 9 in the appendix of the PEARL paper [3], which shows all methods including RL2, MAML and ProMP trained to convergence. These on-policy methods do not reach the asymptotic performance of PEARL, which MIER matches on in-distribution test tasks.
>
> **"Updating $\phi$ only is not a consistent learning algorithm”**
>
> At test-time, after first adapting phi, we also adapt theta, and thus the adaptation procedure is indeed consistent (as described in appendix C.2 ). As the reviewer suggests, it might well be the case that $\phi$ is not expressive enough to learn a completely new task, but in that case subsequent training of the model will ensure the consistency of the adaptation procedure.
>
> **"You don't need to meta-learn the initialisation of $\phi$ , but you can always set it to to 0 before the inner-loop update”**
>
> This review’s observation is indeed correct and we are already using this formulation in our implementation. We considered this an implementation detail, and hence it wasn’t discussed in the paper. We will add the citation of the relevant prior work where this was introduced [4].
>
>
> References
>
> [1] Stadie, Bradly C., et al. "Some considerations on learning to explore via meta-reinforcement learning."
>
> [2] Gupta, Abhishek, et al. "Meta-reinforcement learning of structured exploration strategies." Advances in Neural Information Processing Systems. 2018.
>
> [3] Rakelly, Kate, et al. "Efficient off-policy meta-reinforcement learning via probabilistic context variables." International conference on machine learning. 2019.
>
> [4] Zintgraf, Luisa, et al. "Fast context adaptation via meta-learning." International Conference on Machine Learning. PMLR, 2019.
>
>
> [5] Finn, Chelsea, and Sergey Levine. "Meta-Learning and Universality: Deep Representations and Gradient Descent can Approximate any Learning Algorithm." International Conference on Learning Representations. 2018.

---

### Official Review · AnonReviewer4 · 2020-10-30
**Nice algorithm but underwhelming results**

**Rating:** 4
**Confidence:** 4

**Review:**

- Summary: this paper proposes a novel meta-RL method that works as a 2-stage pipeline (with the 2nd stage being optional). The first stage does model identification by having a network doing meta-SL on a learned model that predicts transitions and rewards. This model uses a MAML-like algorithm to adapt a context variable that is passed to a policy network. This policy network does regular RL taking the context variable (describing the task) and the states as inputs. This approach alone would work well in regular meta-RL, but we aim at robustness under distribution shift, for which we need the second part: experience relabeling. There we use the adapted model from the first part to relabel the data coming from other tasks in meta-training and continue training the policy on that synthetic data.
- Pros:
    - 1. The method is nice, simple and well-founded.
    - 2. We're turning a meta-RL problem into a meta-SL + RL problem; that's a nice decomposition.
    - 3. Generalizing outside of the meta-distribution is useful and making the approach consistent is a good requirement
    - 4. The paper is well-written and clear and the experiments are well-executed.
- Cons:
    - 1. I think the experience relabeling shouldn't work well whenever the state distribution changes substantially and thus we haven't seen the states at meta-training time. For instance, in the ant domain where we go to a corner that has never been explored, there shouldn't be much states to relabel in that portion of the state-space (the experimental results in that particular experiment are also worse than MQL). That defeats the extrapolation purpose of the relabeling and should be discussed further. This issue needs to be discussed further. W.r.t. this point, at the end of section 4 it is claimed that MQL will perform worse when the data support is different because MQL assumes it's the same support, which feels similar to what I'm saying; however, then I don't understand why MQL performs better in Ant-Dir (the example domain I gave).
    - 2. The paper mentions the proposed approach being composed of "two independent novel concepts", but each has been studied before: model identification is common in controls and experience relabeling has the same feeling as Hindsight Experience Replay ` https://arxiv.org/abs/1707.01495 ` or the off-policy corrections of this work ` https://arxiv.org/pdf/1805.08296.pdf .`
    - 3. The experimental results are underwhelming: in the in-distribution the results are roughly on par with PEARL and MQL; which is to be expected since that's not the main contribution of the paper. However, the out-of-distribution tasks results are quite underwhelming, given that this is the only algorithm with out-of-distribution in mind: the proposed approach is only significantly better than MQL in 2/5 and statistically worse in 1 and also only better in 1/5 than its ablation MIER-wR.
- Clarity: high
- Significance: medium
- Originality: medium-low
- Questions:
    - Negated joints means that putting 'a' in that joint is the same as putting -a if the joint was not negated?
    - My understanding is that we are _not_ backproping the policy training back to $\phi_T$ and then back to the model inner loop training that produced $\phi_T$. Is that correct? If we don't, wouldn't it make more sense to do so?
- Note:
    - The explanation of appendix A was useful, thanks for including it.
- Reasoning behind decision: this paper proposes combining two relatively well-established ideas in a novel and potentially-useful way. If the extrapolation experiments were more convincing I would lean towards acceptance; but, as this is currently not the case, I am hesitant to endorse it.

=============
UPDATE POST DISCUSSION

Thank you for your responses.
- I am more convinced about the novelty of the proposed methods.
- I am still unconvinced about relabeling being the best we can do for out-of-distribution state distributions. Building learned methods that generalize more broadly or many-inner-step meta-learners could help quite a bit.
- **I got concerned that a question by AnonReviewer2 "Why for in-distribution experiments, Experience Relabeling was removed?" went unanswered.** Having evaluated MIER-wR, adding MIER to the plots during rebuttal should have been very easy. Moreover, in practice, we shouldn't be able to leverage the fact that we know a task is in-distribution or out-of-distribution and choose between MIER and MIER-wR accordingly (which would also be a bit ugly). Therefore, in my opinion, putting MIER-wR as a representative of the paper instead of MIER is not valid.

Since most of my concerns pre-discussion are still there and we cannot assess whether MIER does well on in-distribution (and MIER-wR is not better than MQL for OOD tasks) I, unfortunately, have to decrease my score from 5 to 4.

- This prompted me to take a closer look at the actual numbers of the plots and I got more concerned. I didn't take it into account in my rating update because I should have brought this up before the discussion:
   1. I feel something is off with PEARL vs MIER-wR when looking at in-distribution vs OOD in half-cheetah vel. In particular, we see PEARL does considerably better than MIER-wR on in-distribution (Fig 2), but its performance starts already much lower than MIER-wR in Fig 6 left.
   2. Another concern is whether the numbers for OOD are "decent" in the sense of creating meaningful policies. Half-cheetah-vel-hard has a reward of ~-325, which is lower than MQL with 0 data for in-distribution half-cheetah-vel. This may be because the evaluated tasks are different, but raises concerns that maybe all baselines and MIER are all doing poorly.

---

> ### Author Response · Authors · 2020-11-21
> **Author Response: Novelty and Effectiveness of MIER (Part 1)**
>
> We thank the reviewer for the detailed comments and constructive feedback. We first summarize our response, and then address the concerns the reviewer has and answer the detailed questions.
>
> The reviewer’s concerns are mostly about the novelty of the method and the effectiveness of the experience relabeling process. The novelty of our methods lies in the decoupling of the meta-adaptation and policy training by meta-training a context-based model with MAML and training a context-conditioned policy. The model can also be used to further improve the policy by relabeling data from other tasks. This formulation effectively turns the meta-RL problem into a meta-supervised learning problem and a regular RL problem, which allows MIER to be trained efficiently.
>
> The effectiveness of our method can be seen empirically on the out-of-distribution experiments. In 4 out of 5 out-of-distribution environments, our method outperforms prior methods. The results indicate that experience relabeling is indeed important for extrapolating to out-of-distribution tasks.
>
> **"The paper mentions the proposed approach being composed of "two independent novel concepts", but each has been studied before."**
>
> The relabeling process of MIER is unrelated to the relabeling method used in goal conditioned RL, and the use of the word “relabeling” is merely a coincidence. In goal conditioned RL, a goal is appended to every transition pair and a relabeling mechanism is used to select the goal for training. The meta-RL problem of this paper doesn’t involve any explicit goal. The relabeling process in MIER generates the synthetic next states and rewards for a new task during adaptation to continue training the policy.
>
> For model identification, the novelty of our method does not come from the use of model based RL, which is a well known concept. Instead, the novelty of our method lies in the adaptation of the context variable through model learning and then conditioning the policy on it. This formulation allows us to decouple a meta-RL problem into a meta-supervised learning problem and a regular RL problem, avoiding the complexity of meta-training the policy.
>
> **"I think the experience relabeling shouldn't work well whenever the state distribution changes substantially"**
>
> It is certainly true that experience relabeling cannot help for arbitrary out-of-distribution tasks. However, previous work in supervised learning ([1]) shows that gradient-based meta-learning methods extrapolate to out-of-distribution tasks better than other encoder-based methods. Therefore, by leveraging the advantage of gradient-based meta-learning method, experience relabeling could help the policy to extrapolate better to out-of-distribution tasks compared to encoder-based methods.
>
> Regarding the data distribution support of tasks, we want to clarify that the support is defined on the joint distribution of transition tuples (state, action, reward and next state). Indeed as the reviewer suggests, if we have completely unseen states there can be no guarantee for the model to extrapolate, and this is true for any meta-RL method since the dynamics and reward function can be arbitrary on unseen states. However, when the states and actions are seen before but the rewards and next states are different in the new task, the data distributions of two tasks still have non-overlapping support, but the relabeling process of MIER is capable of adapting to the new task in this case, while theoretically the importance re-weighting method of MQL cannot.
>
> For the Ant-Dir results, as described previously, the ant is moving into completely new states unseen before and there would be no guarantee on performance for any meta-RL method. We hypothesize that the observed performance difference is a result of network architecture and underlying RL algorithm.

---

> > ### Author Response · Authors · 2020-11-21
> > **Author Response: Novelty and Effectiveness of MIER (Part 2)**
> >
> > **"Negated joints means that putting 'a' in that joint is the same as putting -a if the joint was not negated?"**
> >
> > Yes, the action output of the agent on that joint is negated during simulation.
> >
> > **"My understanding is that we are not backproping the policy training back to $\phi_T$ and then back to the model inner loop training that produced $\phi_T$. Is that correct? If we don't, wouldn't it make more sense to do so?"**
> >
> > The reviewer’s interpretation that we are not backproping the context from the RL agent to the model is correct, and this formulation is by design. If the model is able to adapt successfully, the adapted context contains sufficient information to fully identify the task, effectively turning the meta-RL problem into a regular MDP for the policy. Therefore the context adaptation process does not need training signals from the RL agent. By decoupling the meta-RL problem into a meta-supervised learning problem and a regular RL problem, our method can directly apply off-the-shelf meta-supervised learning and off-policy RL methods for efficient and stable training.
> >
> >
> > References
> >
> > [1] Finn, Chelsea, and Sergey Levine. "Meta-Learning and Universality: Deep Representations and Gradient Descent can Approximate any Learning Algorithm." International Conference on Learning Representations. 2018.

---

> > > ### Comment · AnonReviewer4 · 2020-11-23
> > > **Thanks for your response**
> > >
> > > Thanks for your detailed response. I think I understood everything in your reply so I don't have further questions during this discussion period of the review process. In a few days, I will come back and carefully go through all reviews&responses and update my review.

---

### Official Review · AnonReviewer2 · 2020-11-02
**my review**

**Rating:** 5
**Confidence:** 5

**Review:**

This paper proposes a new meta-RL algorithm containing two main components: 1. Model Identification 2. Experience Relabeling. Model Identification models the next state and reward function condition on the previous state, action, and context.  Experience Relabeling module uses the model learned in the previous step to generate synthetic transitions to improve policy learning. Results on in-distribution tasks are comparable with the previous method and it shows good performance on the out-of-distribution task.

1. Paper claims on page 4 second paragraph that the main difference between context variable in this paper and previous works is that gradient descent is used to adapt the context. That is not entirely true as previous work like MQL does use gradient descent to adapt the context (MQL should be cited there too). Am I missing something here?

2. On page 3, the second paragraph from the bottom, you claim that "a sufficient condition for identifying the task is to learn the transition dynamics and the reward function, and this is exactly what model-based RL methods do." If that is true, why did you use context? I don't think this method will work without context (an ablation study would have been nice with/without context).

3. I am not sure if I fully follow your argument about gradient-based meta-learning with value-based in Appendix A or at least it is a very weak argument about the relation between Q function convergence, bellman backup iteration used in actor-critic, and gradient descent.  Can you clarify a bit?

4. Can you explain how Experience Relabeling helps with out-of-distribution tasks? As far as I understood from your paper, model(\hat(p)) is trained using **current** tasks distribution and this model will be then used to generate synthetic data. Then how come it can help with out-of-distribution tasks when it only saw in-distribution tasks?

5. Why for in-distribution experiments, Experience Relabeling was removed? experience relabeling should be as useful as for in-distribution tasks as for out-distribution ones.

6. Do you **only** update context parameters in Eq. 1? not \theta? if that is true, figure 1 is very confusing as it implies the model( \theta) and context will be updated together.

7. How context is constructed? is it a recurrent network? or?

8. Is it really required to use MAML regression to learn the model? wouldn't be enough to just use simple regression to learn the model?

9. This paper writing needs improvement especially sections 3 and 4 as they don't have a smooth flow. It made me read these two sections multiple times to understand this MIER.  And introduction needs improvement too.

10. Utilizing other benchmarks like meta-world [https://meta-world.github.io/] will be more useful than the current benchmarks to evaluate this method per MQL findings.

11. The paper would have been stronger, if the authors had provided ablation studies in which they closely study different component of their method (w/wo context, simple regression to build model, show who close or far generated data from actual out-of-distribution tasks, etc.)

12. Results in this paper are neither that convincing nor that strong. This method only shows good performance on Cheetah-Negated-Joints and Ant-Negated-Joints on out-distribution.  In addition, I don't understand why this method can't have better performance on in-distribution tasks.  Model + context should be more expressive than just using context ( e.g. MQL vs. this method.)


------- Update after rebuttal ---------

Thanks for your response. Even though your responses clarified some of my comments, I still don't understand how Experience Relabeling can help with OOD and why your method doesn't good enough with in-dist data. As as result, I stay with my current evaluation and score.

---

> ### Author Response · Authors · 2020-11-21
> **Author Response: Context Adaptation, MQL and Extrapolation (Part 1)**
>
> First of all we want to thank the reviewer for the detailed comments and constructive feedback. We first briefly summarize our response and address the reviewer’s concerns below.
>
> The reviewer’s concerns lie mostly in the formulation and necessity of the context adaptation in our method. We want to clarify that the context in our method is simply a learnable weight vector of the dynamics and reward model, and the weight vector is meta-trained using MAML to allow the model to rapidly adapt to new tasks by doing gradient descent on the context. Therefore, the adapted context contains all the information required to identify the task, and hence can be directly passed to the policy. Since the context adaptation process is trained to capture all the information about the task, we do not need to backpropagate from the policy into the context.
>
>
> **"How context is constructed?"**
> **"If that is true, why did you use context? I don't think this method will work without context"**
>
>  The context in our method differs from the context in previous methods since it is not the output of neural network computation. Instead, our context is merely a learnable weight vector of the dynamics and reward model that is meta-trained using MAML.
>
> We need the context in order to pass the task information identified by the meta-trained model to the policy. The context enables us to train a policy for all tasks with any off-the-shelf off-policy RL method (we use SAC in particular) by conditioning on it. By avoiding the direct meta-training of the policy, our method can benefit from the stability and sample-efficiency of regular off-policy RL methods. Without the context, the policy would not be able to receive information about the task and therefore cannot adapt to any particular task. We will add an ablation study for our method without context adaptation in the final version of the paper.
>
> **"Do you only update context parameters in Eq. 1?"**
>
> Yes, we only update the context variable in equation 1. The process of adapting the context variable is also explained under Figure 1.
>
> **"Is it really required to use MAML regression to learn the model?"**
>
> We are tackling the problem of meta-reinforcement learning, so the model needs to be able to adapt to a new task by only observing a few trajectories. Regular supervised learning methods (fine-tuned on new data) cannot adapt well with only a small amount of data ([3]).
>
> **"MQL does use gradient descent to adapt the context"**
>
> According to the description in the MQL paper (see highlighted Remark 2)([1]), the context is built using “an off-the-shelf model like GRU”. So the context in MQL is computed from an GRU encoder. Although MQL does use gradient descent to continue improving the policy like our method does in the experience relabeling process, MQL is never meta-trained with a gradient-based meta-learning approach like the one used in our method.
>
>
> **"I am not sure if I fully follow your argument about gradient-based meta-learning with value-based in Appendix A"**
>
> The core idea of Appendix A is that Bellman backup is not compatible with the MAML objective, making it difficult to combine a value based RL method with MAML. The MAML objective trains a model to adapt to a new task by taking a few gradient steps. However, the Bellman backup can only propagate information of the environment by one time step in the backward direction. This means that it needs at least the same number of steps as the horizon of the task to converge, which is incompatible with the MAML objective. Concretely, consider an example MDP where the agent only receives rewards after 100 steps from the initial state. In this MDP, to fully propagate the reward information into the values of the initial state, we need at least 100 steps of Bellman backup, which cannot be done by MAML.
>
> Our method avoids this problem by decoupling the fast adaptation of MAML with off-policy reinforcement learning. By training a context conditioned model, we can leverage the advantage of gradient-based meta-learning to capture all the information in the context variable. The policy can then be trained via any off-policy RL method by simply conditioning on the context.
>
> **"Can you explain how Experience Relabeling helps with out-of-distribution tasks?"**
>
> Previous work in supervised learning ([2]) shows that gradient-based meta-learning methods extrapolate to out-of-distribution tasks better than other encoder-based methods. Therefore by leveraging the advantage of gradient-based meta-learning methods, experience relabeling could help the policy to extrapolate better to out-of-distribution tasks compared to encoder-based methods, as the experiment results demonstrate.

---

### Decision · Program_Chairs · 2021-01-07
**Final Decision**

**Decision:**

Reject

**Comment:**

The paper was evaluated by 4 knowledgeable reviewers and got mixed scores. While most reviewers appreciated the new intuitive approach to meta RL. there were severe concerns about algorithmic choices and the evaluations that led to a poor score from some reviewers. These concerns are summarized below:
- The motivation of experience relabeling for out of distribution samples is not clear (R2)
- It is unclear why experience relabeling does not work for in distribution samples (R2, R4)
- The reported performance is not a fair comparison as it is typically not known when a task is in-distribution or out-distribution, so we would either have to take always experience relabeling or never (or learn when do use which algorithm)
- The paper falls short in terms of evaluations (R3, R4), in particular it remains unclear to me if MIER can, under realistic circumstances. It is suggested to use more established benchmarks such as Meta-World to evaluate the performance of MIER.

For the given reasons, I recommend that the authors do these corrections and  go through another round of reviews at another conference.